# On the Fabrication and Characterization of Polymer-Based Waveguide Probes for Use in Future Optical Cochlear Implants

**DOI:** 10.3390/ma16010106

**Published:** 2022-12-22

**Authors:** Christian Helke, Markus Reinhardt, Markus Arnold, Falk Schwenzer, Micha Haase, Matthias Wachs, Christian Goßler, Jonathan Götz, Daniel Keppeler, Bettina Wolf, Jannis Schaeper, Tim Salditt, Tobias Moser, Ulrich Theodor Schwarz, Danny Reuter

**Affiliations:** 1Fraunhofer Institute for Electronic Nanosystems ENAS, 09126 Chemnitz, Germany; 2Center for Microtechnologies (ZfM), Technical University of Chemnitz, 09126 Chemnitz, Germany; 3Experimental Sensor Science, Technical University of Chemnitz, 09126 Chemnitz, Germany; 4Institute for Auditory Neuroscience and InnerEarLab, University Medical Center Goettingen, 37075 Goettingen, Germany; 5Institute for X-ray Physics, University of Goettingen, 37075 Goettingen, Germany; 6Multiscale Bioimaging Cluster of Excellence, University Medical Center Goettingen, 37075 Goettingen, Germany

**Keywords:** SU-8, PMMA, polymer, waveguide, optical cochlear implant, spin coating, etching

## Abstract

Improved hearing restoration by cochlear implants (CI) is expected by optical cochlear implants (oCI) exciting optogenetically modified spiral ganglion neurons (SGNs) via an optical pulse generated outside the cochlea. The pulse is guided to the SGNs inside the cochlea via flexible polymer-based waveguide probes. The fabrication of these waveguide probes is realized by using 6” wafer-level micromachining processes, including lithography processes such as spin-coating cladding layers and a waveguide layer in between and etch processes for structuring the waveguide layer. Further adhesion layers and metal layers for laser diode (LD) bonding and light-outcoupling structures are also integrated in this waveguide process flow. Optical microscope and SEM images revealed that the majority of the waveguides are sufficiently smooth to guide light with low intensity loss. By coupling light into the waveguides and detecting the outcoupled light from the waveguide, we distinguished intensity losses caused by bending the waveguide and outcoupling. The probes were used in first modules called single-beam guides (SBGs) based on a waveguide probe, a ball lens and an LD. Finally, these SBGs were tested in animal models for proof-of-concept implantation experiments.

## 1. Introduction

With more than a million users currently worldwide [1], electrical cochlear implants (eCI) are most likely the most implanted neuroprosthesis. An eCI stimulates spiral ganglion neurons (SGNs) by means of an electrical pulse, and thus bypasses the defect of lost hair cells of a deaf person. eCIs partially restore hearing in most users, yet its performance is limited due to a broad spread of the electric field surrounding the electrode and thus stimulating a large amount of SGNs, which results in an impairment of the frequency resolution [2,3]. In addition, the intensity coding is rather poor concerning a small output dynamic range [4,5]. Based on those limitations, it is difficult for eCI patients to understand speech in background noise [6]. A solution could be the use of light instead of electrical pulses, as light can be focused much better. An optical cochlear implant (oCI) that stimulates the SGNs by means of optical pulses could improve hearing significantly by stimulating fewer SGNs with a single pulse. This leads to a higher-frequency resolution [7,8,9,10]. However, SGNs are not light-sensitive per se. The control of cells by light envisioned decades ago [11] by optogenetics combines optics and genetics. To achieve this, light-gated ion channels such as channelrhodopsin-2 (ChR2) are genetically introduced [12], which have a maximum sensitivity in the range between 460 and 480 nm [13]. To investigate the feasibility of oCIs, implantable 230-µm-wide flexible µLED probes were fabricated, combining two wafer-level processes and the layer transfer of the LEDs from the sapphire growth substrate to a carrier wafer via metal wafer bonding and laser lift-off [10,14,15].

In this work, another approach is pursued by fabricating flexible waveguide probes, which guide light from an external source to the cochlear. Two polymers with different refractive indices are needed for ensuring both total internal reflection between core and cladding layer and device flexibility [16], in contrast to rigid waveguide probes with polymer core and silicon oxide-based cladding layers [17,18]. Waveguides exhibit bending [19,20,21] and scattering [22,23] losses, which are experimentally investigated in this paper. Total losses in SU-8 waveguides were reported to be in the range of 0.2 dB/cm up to 6.4 dB/cm [24,25,26,27]. A material combination of PMMA/SU-8/PMMA is deposited for the realization of the flexible polymer-based waveguides. To structure the polymer-based layer stack, the PMMA/SU-8/PMMA is patterned by UV lithography and dry etching of 6” silicon wafers. The implants are based on optical waveguides which are designed as bendable and flexible devices. Therefore, the polymer structures have to be released from the substrate by sacrificial layer etching after fabrication. Waveguide probes can be used in a wide field of sensor applications. For example, waveguide probes with inscribed Bragg gratings embedded in 3D-printed ABS sensing pads for force measurements were developed [28], as well as sensors for environmental monitoring and drinking water quality control based on a gold-coated tilted fiber Bragg grating with immobilized *Acinetobacter* sp. Bacteria on the gold surface used as receptors [29]. Polymer waveguides are frequently also used as light probes in photonic sensor networks [30] and optogenetics [16,17,18,27,31,32].

## 2. Waveguide Design and Fabrication Process

### 2.1. Design of the Flexible Polymer-Based Waveguides

The layout of the waveguide probes is shown in Figure 1a. It consists of flexible polymer-based waveguides guiding the light from the laser diodes (LD) bonded directly on the polymer material to the light-outcoupling structure that is at the tip of the flexible polymer waveguide probe. A certain length of the probe from the tip is intended for implantation into the cochlea. Different types of these waveguide probes are realized within this development and are arranged quarter-clockwise on a 6” wafer (Figure 1b). Different waveguide probe lengths, a number of waveguide probes and the bending radius of the waveguide probe have been designed to observe the influence of outcoupling and bending losses as well as the control of the individual waveguide probes within a ten-channel waveguide system. The waveguide probe shown in Figure 1a is characterized by ten individual, directly controllable waveguides, a probe length of 45 mm and 5 µm × 5 µm outcoupling structures. The detailed cross section of the polymer waveguide probe system is explained within its fabrication process in Section 2.2.

### 2.2. Wafer-Level Fabrication of the Flexible Polymer-Based Waveguides

The wafer-level fabrication of the flexible polymer waveguides is shown in detail as schematic in Figure 2 and is separated in three columns by means of the status of the following technology flow as cross-section views of (I) the outcoupling structure, (II) the waveguide probe and (III) the side view of the LD coupling structure. The wafer-level fabrication is realized on 6” Si wafers and starts (Figure 2a) with the deposition of a 100 nm thermal SiO_2_ and spin coating of a 1 µm thick sacrificial layer LOR 10 B (from Microchemicals GmbH—MC) used to detach the polymer waveguides from the Si substrate at the end of the fabrication. Furthermore, a 5 µm thick PMMA layer (from Micro Resist Technology GmbH—MRT) acting as supporting layer is used, as well as a 200 nm SU-8 2000.5 (MRT) adhesion layer needed for the bottom cladding PMMA layer, which is realized later by spin coating. The subsequent sputter deposition of the metal layer stack starts with an adhesion layer consisting of 20 nm Cr followed by 100 nm Au seed layer. Afterwards, electroplating is performed for 3 µm thick Au bond pads by negative-tone resist AZ125nXT (MC) with 12 µm thickness. After electroplating the bond pads, the negative-tone resist AZ125nXT (MC) is removed. Due to the 3 µm high Au bond, to remove the Au seed layer and Cr adhesion layer, a lithography step is used with spray coating of 7.5 µm AZ4999 (MC), followed by dry etching to form the bond pads and the light-outcoupling structures at the end of the waveguides (Figure 2b). A supporting PMMA layer that acts in addition as the bottom cladding layer underneath the SU-8 waveguide core layer is used in the following step to adjust the LD emission facet to the core of the waveguide. Therefore, a 2.5 µm thick PMMA (MRT) layer is spin-coated on the wafer followed by a SU-8 2000.5 layer of 200 nm to avoid cracks that occurred in the 2.5 µm bottom cladding PMMA when the following SU-8 3005 (MRT) layer is directly processed on top (Figure 2c). The SU-8 3005 waveguide core layer is spin-coated with a thickness of 5 µm. Afterwards, the SU-8 3005 is exposed with the waveguide-layer mask in a mask aligner MA200 (SUSS MicroTec GmbH) with 100 mJ/cm² for 3 s. Finally, the SU-83005 is developed in mr-Dev 600 (MRT) topside down for 8 min and rinsed 1 min in isopropanol and DI-water. Thus, the waveguide and the outcoupling structures are realized. A 2.5 µm thick PMMA cladding layer is spin-coated on top of the SU-8 3005 waveguide core layer (Figure 2d). Afterwards, a 50 nm thin Ti hard mask is deposited by sputtering and patterned by wet etching in buffered 0.5% hydrofluoric acid using a lithography mask realized by spray-coated AZ4999 resist (Figure 2e). Finally, the polymer layer stack of 2.5 µm top cladding PMMA, 200 nm adhesion layer SU-8 2000.5, 2.5 µm PMMA bottom cladding/supporting pedestal layer, 200 nm adhesion layer SU-8 2000.5 and the 5 µm PMMA substrate layer is patterned by lithography with the waveguide probe system, which is etched using the Ti hard mask in an O_2_ dry etching process at a Sentech SI 500 etch tool with 50 sccm O_2_, 60 W bias and 300 W ICP down to the SiO_2_. Afterwards, the Ti hard mask is removed selectively to the polymer stack in a dry etching process with 36 sccm Cl and 15 sccm Ar at 160 W bias and 500 W ICP (Figure 2f). Finally, the LDs are bonded on the 3 µm thick Au bond pads on the flexible polymer-based waveguide probes (Figure 2g). 

Figure 3a shows the waveguides and a photo of the light-outcoupling structures, as well as a detailed microscope image from the region of interest. Polymer-based waveguide probes and light-outcoupling structures are shown as SEM images in Figure 3b as a cross section. An SEM image as a detailed view of the cross section of the polymer waveguide with its cladding and core layers is given in Figure 3c.

## 3. Waveguide Characterization

The fully processed wafers (Figure 4a) have to be investigated concerning roughness-induced scattering and failures such as scratches and particles enclosed in the different layers or process residues. Each of these failures has the potential to disturb the light guiding properties of the waveguides. Hence, different methods, which will be introduced in the following, are used to identify such failures and optical loss mechanisms.

### 3.1. Bright- and Dark-Field Microscopy

Optical microscope images were used as a first method of identification of failures in the waveguide structures. Some failures—such as particles—are well detectable with bright-field microscopy. Other ones, such as interface roughness, are visible in dark-field microscopy in a better way. Thus, bright-field and dark-field microscopy were used supplementarily. In addition, SEM was used to obtain a more detailed view on specific surface structures, e.g., etching-related damage.

### 3.2. Scattered Light Detection Setup

For scattered light detection, the wafers were separated first into four quarters by the scribe-and-break method. Thus, flat waveguide facets could be achieved, facilitating the incoupling of focused laser light. 

The scattering light setup is shown in Figure 4b and comprises both a probe and laser stage and a camera system to couple in laser light and observe the probe’s scattered light from a top-view position based on a LabView program. The collimated laser beam is focused onto the scribe-and-break-generated waveguide facet by a mounted aspherical lens (A220TM-A, Thorlabs) with 11 mm focal length and a numerical aperture (NA) of 0.26. The lens has a working distance of 6.91 mm and the lens surface is AR-coated for the spectral range from 350 nm to 700 nm. Probes without gold bonding pads were used for these experiments. By maximizing the signal at the outcoupling structure by means of using the two stages, and thus positioning the waveguide’s facet precisely into the focus point of the asphere, the coupling into the waveguide can be optimized. Once an optimum is reached, the software stores pictures of the whole probe at different exposure times to enhance the dynamic range of the camera in order to evaluate the bright outcoupling sides and the comparably dark waveguide shank simultaneously.

For further analyzing the integration time series of images, first, a Mathematica script extracts the relevant waveguide parts. Forming the sum of intensities in one pixel column for all columns in every picture, one-dimensional plots of the intensity along the waveguide shank can be obtained.

### 3.3. Probe Diagnostics

Figure 5 shows different images of an ideally processed probe. In Figure 5a, the tip of a probe in dark-field microscopy is shown. The outcoupling structures can be clearly seen as bright areas at the end of every waveguide, and the waveguide walls are smooth, which is crucial for guiding light with low losses. In Figure 5b, a top-view image of a probe tip is shown when light is coupled into one waveguide. On the right side of the image, the waveguide can be seen. The guided light is slightly scattered at the waveguide walls, which is detected by a camera. Due to back-reflected light and partial crosstalk between waveguides, the outcoupling structures of all waveguides appear bright in the image. The most intensive and overexposed outcoupling structure corresponds to the waveguide where light is coupled in. The detailed analysis of respective light intensities can be found in Figure 6.

The dark-field microscopy image in Figure 5c shows a probe’s tip with missing outcoupling structures. The outcoupling structures are made of SU-8 (MRT), which is a negative-tone resist, and the single elements have a small footprint of 5 µm each. The development time had to be chosen carefully in order to prevent undercut and related detachment of the small structures. Figure 5d shows outcoupling sides without outcoupling structures. A waveguide with incoupled light is visible on the right hand side. However, at the probe tip, there is no dominant outcoupling position concerning light intensity. A rather diffuse region of brighter light intensities is visible instead. This is due to the missing outcoupling structures at the end of the waveguide, which leads to a scattering of the light at different sides along towards the probe tip at the ends of the neighboring waveguides.

In Figure 5e, a dark-field microscopy image of a waveguide and the corresponding outcoupling structures is shown. The upper part of the waveguide appears bright, different to the waveguides in Figure 5a,c. In addition, these waveguides show untypically high intensity losses. A further investigation with an SEM showed that the SU-8 core of these waveguides was unintentionally etched due to nonsufficient thickness of the metal hard mask (Figure 5f).

### 3.4. Scattering Loss Analysis

The graph shown in Figure 6 serves as basis for the evaluation of losses caused by scattering. By applying a fit to the scattered light graph in the region of the shank of one probe’s waveguide (red, violet and orange bars in Figure 6), a slope can be determined. This slope corresponds to a scattered light attenuation parameter *S*, describing the fraction of the light getting lost per distance traveled through a waveguide.

The scattered light curve in Figure 6 reveals higher values at the left side in a region of up to 1 cm after the waveguide facet. Some light of the LD is coupled into the cladding and damped fast in the region close to the entrance facet. On the right side of the curve, ten local maxima are detected, indicating the outcoupling structures of the ten waveguide channels. The most intense one is the fifth one, which corresponds to the position of the outcoupling structure of the addressed waveguide. Accordingly, the other maxima correspond to the outcoupling structures of the other nine waveguides, which is caused partially by crosstalk of the waveguides and back-reflection of intentionally outcoupled light. Compared to the other nine maxima, the most intense one is between 12 dB to 22 dB brighter. The addressed outcoupling site is about 33 dB brighter than the middle region of the waveguide shank. The low light intensity in this region indicates that light is guided with low intensity loss. The intensity loss in this straight-waveguide region is exclusively caused by scattering. A fit was applied to the scattered light curve in that region to extract a scattered light attenuation parameter from the curve. However due to surface-quality-related leaps, the curve does not show a broad region along the probe’s shank where the application of one linear fit is applicable. Thus, three narrower regions without leaps were identified to apply separate linear fits (red, violet and orange bars in Figure 6). In these regions, scattered light attenuation parameters of 1.04 dB/cm, 0.78 dB/cm and 0.30 dB/cm could be derived from the red, violet and green fitting curves, respectively. These parameters are reproducible for other waveguides on the same wafer and represent a range of waveguide qualities.

### 3.5. Bending Loss Analysis

Figure 7 shows the measured scatter light for a probe with double bends of 0.1 cm radius each at a longitudinal pitch of 5 mm. A locally increasing detected light intensity is caused by light more efficiently redirected into the objective of the measurement camera. An on-average steeper decrease in the intensity in the region of the bends can be observed, indicating additional bending losses, which will be examined in detail later.

Optical microscope images were used to calculate the contribution of scattering losses and bending losses to the total loss of a bended waveguide. First, the total length *l* of the waveguide has to be determined. The length of the waveguide without any bends is *d*. Depending on the number of bends *N* and the additional length Δ*s* caused by one bend, the total length *l* of a waveguide can be calculated as
(1)l=d+N·Δs
with
(2)Δs=s−Δd.

The straight distance between the starting point and the end point of the bend is Δ*d*, which is known from the design of the lithography mask. The distance along one bend
(3)$$s=\frac{4\theta }{360}\cdot u$$
where *u* is the periphery of a circle, with bending radius *r* and *θ* the angle per bend.

To extract the total loss from an intensity graph of a test structure, a linear fit is applied between two different maxima of the graph, as shown in Figure 7. The maxima are caused by several turnings on the test structures. With the knowledge of the positions P1 and P2 of the maxima and the intensities at these points, which can be read directly from the graph, the straight distance *d* and the total intensity loss between these two points can be identified. Using Equations (1)–(3), the total distance *d* that the light has to travel through the waveguide can be calculated. Total intensity losses can be determined by multiplying this distance with the scattering loss of straight waveguides per length. The remaining losses are bending losses. To make bending losses of different bending radii comparable to each other, the bending losses are related to 360°.
(4)$$\alpha_{\mathrm{bend},\;360{}^{\circ}}=\frac{\alpha_{\mathrm{bend}}}{4N\cdot \theta }\cdot 360{}^{\circ}$$

In addition, the scattering losses can be related to a full circle of 360°.
(5)$$\alpha_{\mathrm{scat},\;360{}^{\circ}}=2\pi \cdot r\cdot 0.78\;\frac{dB}{cm}$$

Based on the scattered light attenuation parameters from Section 3.4, the total loss and the contributions of scattering and bending losses can be calculated according to the procedure described above. Structure parameters necessary for this calculation are shown in Figure 7. The results from the calculation can be found in Table 1.

The bending losses are expected to increase exponentially with decreasing bending radius as the angle under which light is been reflected at the inner walls of the waveguide approaches the angle of total reflection, especially for higher-order lateral modes. This effect can clearly be found in the data in Table 1. Bending losses become the dominant path of losses for decreasing bending radius. Due to exponentially increasing bending losses, total losses per 360° increase with the decreasing bending radius. The determination of three attenuation parameters permits the estimation of the total losses of bent waveguides. The dependencies of the different losses on the bending radius and the corresponding ranges are shown in Figure 8. Bending losses are dominant for bending radii up to 2.5 mm and likely beyond. Therefore, it is advantageous to minimize the bending radius in the tapered section at the initial part of the probe (compare Figure 1a) to reduce overall losses. The measured bending losses are for bends in the plane of the probe. Out-of-plane bending losses due to the curvature of the probe when following the geometry inside the cochlea can be estimated from these data, but should be evaluated separately.

## 4. Application of Waveguide Probes

### 4.1. Single-Channel Module Assembly

A procedure for the preparation of modules was established the so-called single-beam guides (SBGs) for first tests of the waveguides in animal models. They comprise an LD as light source; a 3D-printed ABS (acrylonitrile butadiene styrene) shell with a ball lens fixed inside for focusing the light; another 3D-printed holder with the waveguide probe fixed inside; and the LD electrical connections, including an ESD protection diode.

For the preparation of an SBG, two parts have to be set up, which are assembled at a later stage. The preparation of the first part, the laser module, starts with the 3D printing of the ball lens shell. For printing, an Ultimaker 3 FDM printer was used.

A 2 mm diameter sapphire ball lens was fixed with black epoxy (E320, Epoxy Technology) inside the ball lens shell. As the shell defines the position of the ball lens in front of the laser facet and thus the image distance, the correct ball lens position is crucial for efficient coupling of the laser light. Therefore, the upper part of the shell is pressed to a smooth surface (e.g., an object slide) to allow for a reproducible positioning of the lens inside the shell (Figure 9). After mounting the lens, the black epoxy was applied and cured for 2 h at 65 °C in an oven.

After applying and curing the transparent epoxy filling the socket for 2 h at 65 °C, the socket was screwed in a lens tube (SM05L10, Thorlabs). The rear side was closed with a cap (SM05CP1, Thorlabs) with a drilled hole inside to mount the SMA jack. Finally, transparent epoxy is applied to the thread mounting the laser socket in the lens tube to prevent relative motions of the thread. The transparent epoxy is cured for 2 h at 65 °C. No active alignment is necessary up to here. All measures are made to warrant the best possible reproducibility of the geometry. The waveguide probe module preparation starts with 3D-printing a waveguide probe holder shell and another part for holding the probe and encapsulation, respectively. At this point, the waveguide probe is still on the wafer, which serves as substrate. However, to facilitate an implantation into a cochlea, the probe has to be flexible. Indeed, incoupling tests showed that coupling light into a fully detached waveguide without Si substrate at the incoupling side is very difficult. Incoupling becomes more stable and feasible for probes with remaining Si substrate underneath the waveguide facets. As shown in Figure 10a, the probe is partly immersed upside-down into a developer solution (ma-D 533, MRT), which dissolves the sacrificial layer LOR 10 B (MC) between the probe and the Si substrate, and thus removes the probe from the Si substrate. After dissolving the sacrificial layer, the released part of the Si substrate is removed by the scribe-and-break procedure, scribing the edge of the silicon part at the front side (Figure 10b).

After gluing the Si substrate to the probe holder with transparent epoxy and curing it for 2 h at 65 °C, the holder is encapsulated with a 3D-printed light-blocking part. Existing joints are closed with black epoxy, hindering light leakage. After this step, the probe module is ready to be assembled together with the LD module.

For assembling the LD module and the probe module, the two parts were fixed on an optical table and a manual 5-axis stage, respectively. The waveguide probe was observed with the top-view camera system introduced in Chapter 3.2. After adjusting the laser beam orientation along the optical axis, the adjustment of the probe module in front of the LD module was performed. Once the highest intensity value at the outcoupling point of the waveguide is detected, the probe stage axis was driven back along the optical axis. The transparent epoxy was stirred for about 30 min on a hot plate at 50 °C and 1500 rpm to catalyze the epoxy cross-linking process and increase its viscosity, facilitating the application to the glued surface. Afterwards, the epoxy was applied to the bottom side of the probe module. The adjustment process was finalized by positioning the probe stage axis again at the predetermined ideal distance in respect to the LD module. All the active alignment steps in the second part of the procedure can in principle be automated to allow for production in larger quantities. A schematic side view of the aligned laser and waveguide probe module can be seen in Figure 11.

After curing the epoxy for 24 h at room temperature, the SBG can be released from the optical alignment setup. An example of an operated SBG module is depicted in Figure 12. The output power of the SBG was measured by inserting the waveguide probe into an integrating sphere. In addition, long-term tests of not electrically driven SBGs were conducted by immersing the polymer waveguide into buffered saline solution at 37 °C. At temporal intervals the probe was removed from the liquid and inserted into an integrating sphere to remeasure the output power of the SBG.

### 4.2. Module Degradation Tests

As described at the end of Section 4.1., the SBGs were subjected a lifetime test where the optical output power of the SBGs was measured frequently. The temporal evolution of the output power of four SBG modules is shown in Figure 13.

With about 13 mW output power, SBG013 exhibits the highest optical output power. At an operating current of 92 mA, the output power of the LD used in an SBG in constant wave mode is 40 mW. Hence, the optical efficiency of SBG013 is determined to be 33%. Compared to the other SBGs, SBG013 shows a lower decrease in output power due to minimizing the waveguide aging by not storing the probe in saline solution. Thus, the decrease should be exclusively incoupling-alignment-related. It can be enhanced further by employing more advanced lens-coupling solutions compared to the applied ball lenses, e.g., aspheric lenses correcting spherical aberrations.

In addition, the output power of each of the four SBGs shown in Figure 13 decreases over time. The main reason for that should be a decrease in the quality of incoupling, as it is spatially very sensitive. One reason for that could be the creep behavior of the epoxy used for fixing the probe shell with the probe in front of the lens shell. Another reason could be a degradation of the 3D-printed shells themselves due to UV exposure from sun or luminescent tubes, because the ABS used for the shells is not UV-resistant. This could lead to a realignment of the relative positions of the probe and the lens, which influences the incoupling efficiency. In addition, the 3D-printed ABS parts should be replaced by more rigid metal solutions for long-term stable applications, and epoxy gluing should be performed with thin joint distances in the range of 0.1 mm to minimize creeping. Further investigations with and without saline solution are necessary to separate the effects of the degradation of the geometrical incoupling from the influence of the saline solution on the waveguide degradation.

## 5. Cochlear Implantation

Waveguides were manually coated with up to four layers of silicon and inserted into cochleae of gerbils, an animal model commonly used in cochlear implant research, in a surgery situs after euthanasia. The waveguide probe itself only exhibited a thickness of 10–15 µm of polymer materials. Previous work showed that a deep implantation was beneficial for covering a greater area of the hearing range, which led to less compression of the frequency information and a sufficient amount of usable electrodes, which lastly correlated with an increase in speech recognition to a certain extent [33].

The part of the implant that is inserted into the round window should exhibit long-term mechanical stability as well as flexibility to reduce insertion trauma. Yet, to allow for a safe and reproducible implantation, it also needs to have the necessary stiffness [34], causing an interplay of two parameters that needs to be considered. The stiffness is needed to allow for the insertion of the probe without having it bend away from the insertion force, yet at the same time it causes stronger shearing force against the osseous structures of the cochlea and can also cause the implants position to change from the Scala tympani to the Scala vestibuli by piercing the basilary membrane.

In line with known parameters of the gerbil cochlea [35], the mechanical parameters of the probe allowed for successful implantation followed by micro-CT imaging in four out of four probes tested so far, yet the probes varied in insertion depth as well as damage caused in the process of implantation, which will be the subject of further investigation in the near future. The probes are already produced with an intrinsic material memory in a spiral shape due to the fabrication, which is beneficial for an easier implantation but also helps to reduce the distance between the outcoupling structures and the neuronal cells in the modiolus, which consequently also leads to an increased spectral resolution.

The focus of further development will therefore also lie on the mechanical properties of the probe. A graded silicone layering of the waveguide probe with lower stiffness at the probes tip compared to its base, as mentioned by [34], could increase the implantation success. Furthermore, an increase in stiffness perpendicular to the natural bending of the implant could be an option to reduce damage through insertion [36]. Figure 14 shows an XPCT scan of implanted waveguide structures in the cochlea with an implanted length of 11.5 mm at a total cochlea length of 13 mm, resulting in an implanted cochlea fraction of 88%.

## 6. Conclusions

A 6” wafer-level micromachining process for the fabrication of flexible waveguide probes, made of PMMA as cladding and SU-8 as core material, was established. By means of a scattered light detection setup, as well as different microscopy methods such as bright- and dark-field optical and SEM microscopy, fabricated waveguide probes were evaluated, concerning failures and intensity loss in operating mode, i.e., when light was coupled into the single waveguides. A profound evaluation of the scattered light experiment made it possible to determine the contribution of scattering and bending losses to the total intensity loss of the waveguides. In this work, the scattering attenuation parameter was determined to be (0.7 ± 0.3) dB/cm.

The probes were further processed to establish a single-channel prototype system designed for the application in animal models. Those probes exhibit up to 33% optical efficiency at an output power of up to 13 mW.

## Figures and Tables

**Figure 1 materials-16-00106-f001:**
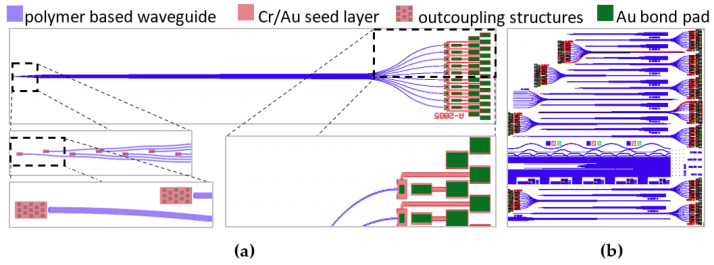
(**a**) Layout of a ten-beam flexible polymer based waveguide probe with details of (left) the LD bond pads at coupling side and (right) the light-outcoupling structures. (**b**) Layout of the 6” wafer with the different types of waveguide probes.

**Figure 2 materials-16-00106-f002:**
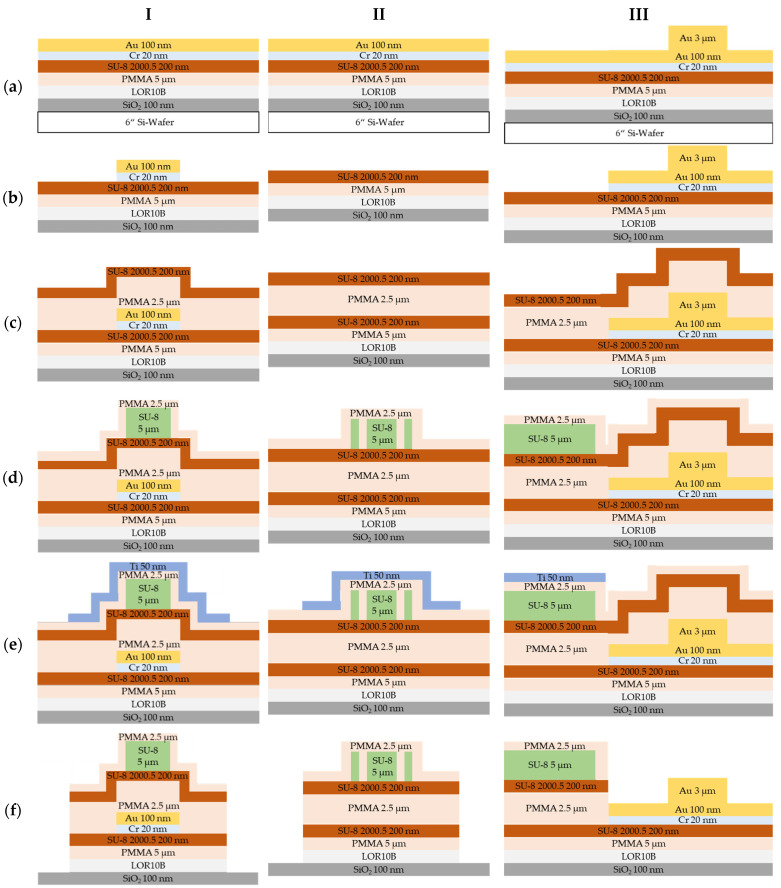
Fabrication sequence of the polymer waveguides as cross views for (I) the light-outcoupling structures, (II) the waveguides and (III) the side view of LD coupling side. The 6” silicon wafer is just shown in (**a**), but is in every step (**b**–**g**) part of the flexible polymer waveguide probe.

**Figure 3 materials-16-00106-f003:**
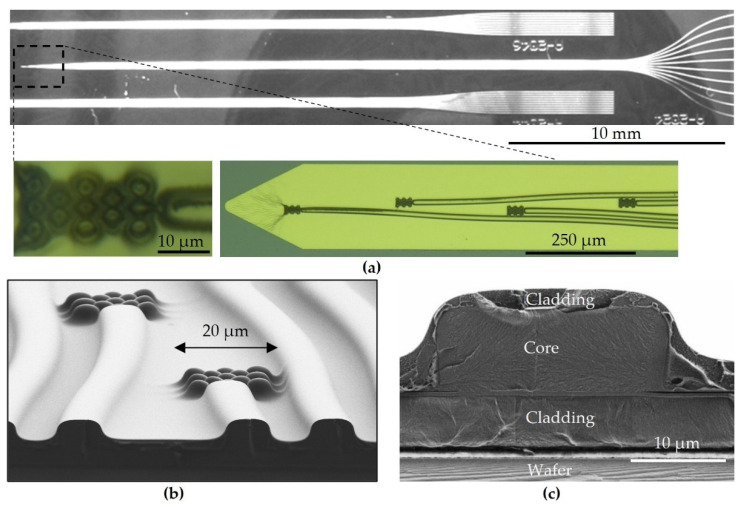
(**a**) Microscopy image of a fabricated polymer-based waveguide probe and details of outcoupling structures. SEM images of the cross section of (**b**) waveguide probe with outcoupling structures and (**c**) waveguide facet after scribe-and-break procedure of silicon wafer.

**Figure 4 materials-16-00106-f004:**
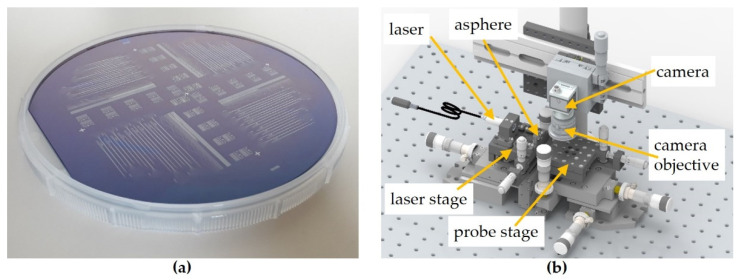
(**a**) Fully processed wafer with different waveguide probes and test structures; (**b**) scheme of experimental setup for scattered light detection.

**Figure 5 materials-16-00106-f005:**
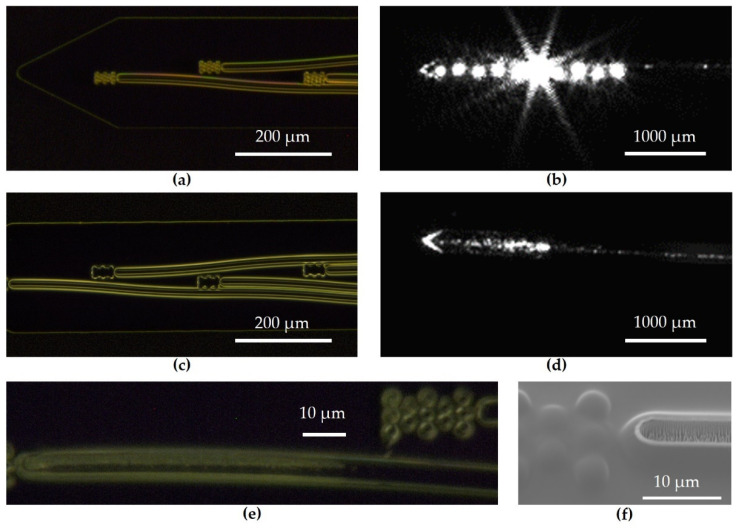
Different microscopy images of waveguides and outcoupling structures of different probes: (**a**) Dark-field microscopy of a probe’s tip with waveguides and outcoupling structures; (**b**) Microscopy image of an operating probe tip; (**c**) dark-field microscopy of a probe tip with missing outcoupling structures; (**d**) microscopy image of an operating probe tip with missing outcoupling structures; (**e**) dark-field microscopy of an etched waveguide; (**f**) SEM image of an unintentionally etched waveguide with outcoupling structure.

**Figure 6 materials-16-00106-f006:**
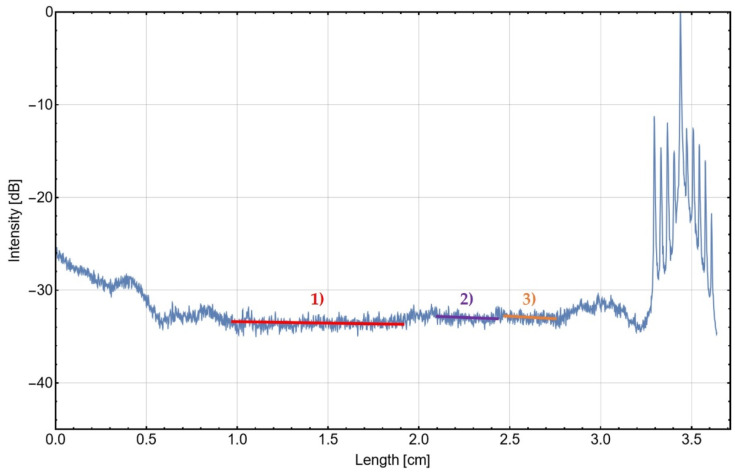
Scattered light intensity along a waveguide probe shank. High values on the right side of the graph indicate the dedicated outcoupling structures of the waveguide. The incoupling causes light scattering at the waveguide’s facet, which influences the graph on the first centimeter after the facet. Three regions 1), 2) and 3) are identified for linear fitting the scattered light detection curve and determining scattered light attenuation parameters.

**Figure 7 materials-16-00106-f007:**
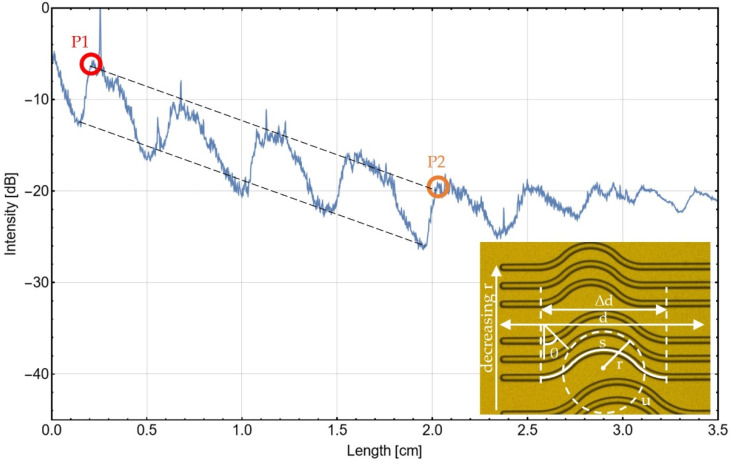
Scattered light graph along four bends from P1 to P2 with radius 0.1 cm of a waveguide. At points P1 and P2, light intensity and position is recorded for further evaluation. Inset: Waveguides on a wafer with different bending radii and other parameters needed for the evaluation of intensity losses along the waveguides.

**Figure 8 materials-16-00106-f008:**
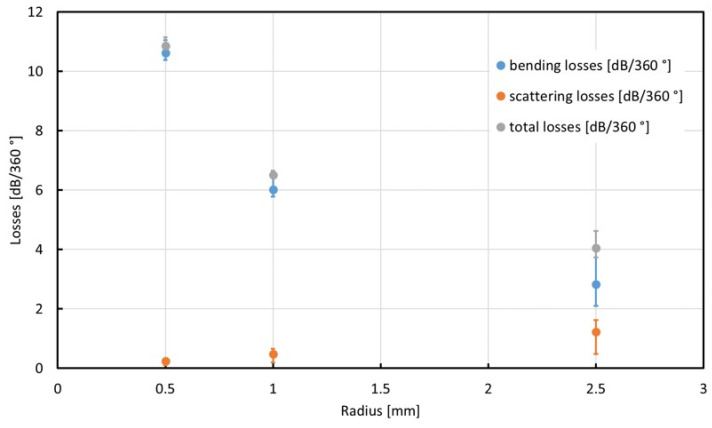
Curvature radius dependencies of bending, scattering and total losses. By means of the determination of three different scattered light attenuation parameters, a range for the losses can be derived.

**Figure 9 materials-16-00106-f009:**
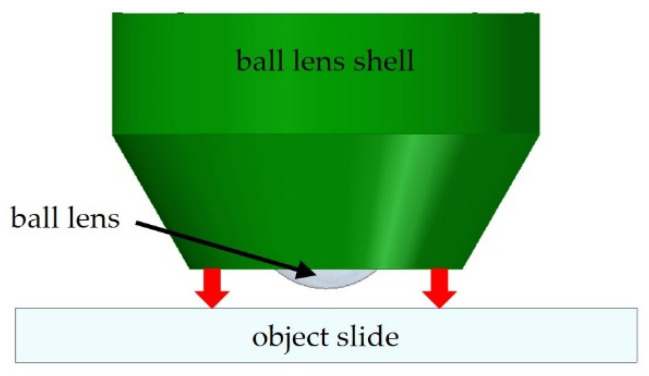
Principle of ball lens positioning inside the ball lens shell.

**Figure 10 materials-16-00106-f010:**
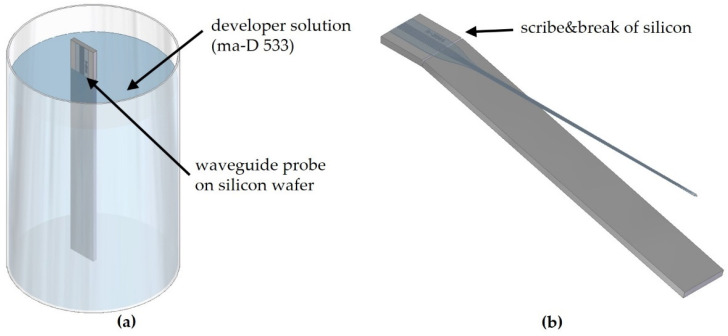
(**a**) Scheme of a waveguide probe immersed upside down in the developer ma-D 533 and (**b**) scribe-and-break procedure to remove surplus silicon from the released probe. To fix the LD in a defined position, a special socket (S05LM56, Thorlabs) is used where the LD is screwed in. The ball lens shell is slid over the LD’s cap and glued to the socket with transparent epoxy (E301, Epoxy Technology). In the next step, the epoxy is cured for 2 h at 65 °C.

**Figure 11 materials-16-00106-f011:**
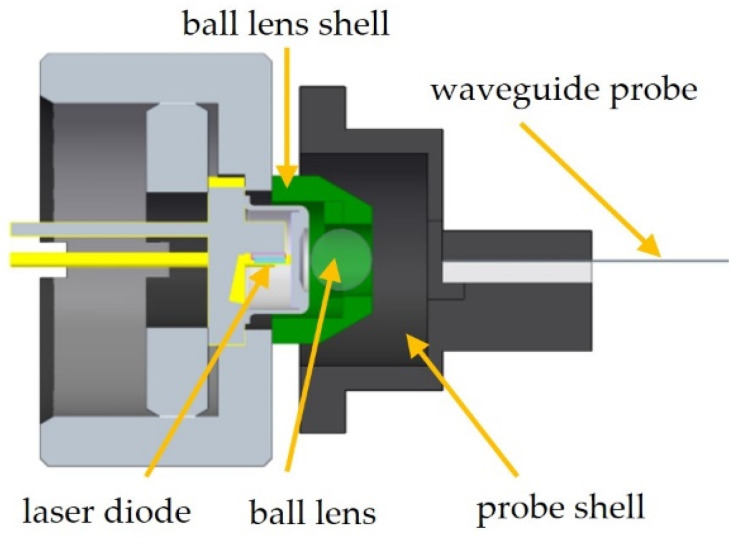
Side view of aligned LD and waveguide probe module. The gap between the two modules is filled with transparent epoxy.

**Figure 12 materials-16-00106-f012:**
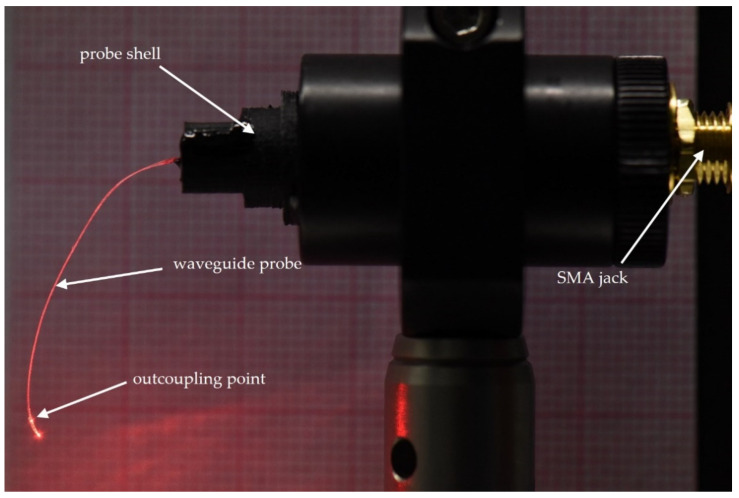
SBG in operation. Via a SMA jack, a source–measure unit (SMU) is connected, which electrically drives the SBG.

**Figure 13 materials-16-00106-f013:**
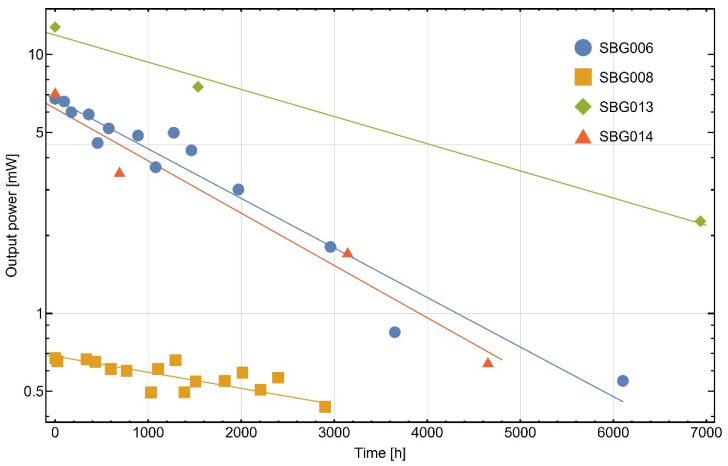
Fitted temporal trend of the output power of four SBG modules. Except for SBG013, probes were stored in buffered saline solution without operating the LDs. After temporal intervals, the probes were removed from the solution and the optical output power of the SBGs was measured in an integrating sphere. R² values of the four fits are 0.94628 (SBG006), 0.72605 (SBG008), 0.99142 (SBG013) and 0.96180 (SBG014).

**Figure 14 materials-16-00106-f014:**
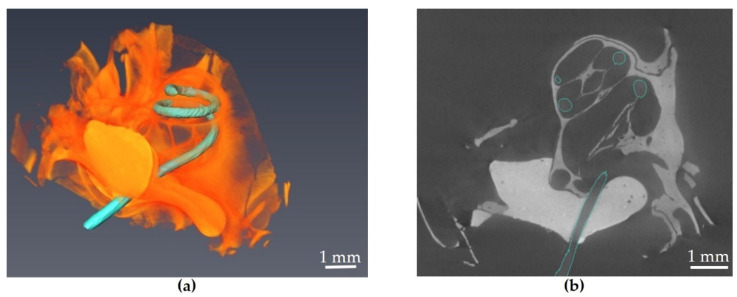
Implanted waveguide probe (marked green) after silicone encapsulation in XPCT scan image, (**a**) 3D reconstruction and (**b**) cross view.

**Table 1 materials-16-00106-t001:** Structure parameters and scattered light measurement results for PMMA/SU-8/PMMA waveguides on a silicon wafer.

Structure Parameters/Measurement Results	Radius 1	Radius 2	Radius 3
	2.5 mm	1 mm	0.5 mm
**Geometrical parameters**			
Bending angle θ [°]	30	45	45
Distance s along one bend [cm]	5.24	3.14	1.57
Straight distance ∆d [cm]	5.0	3.0	1.5
Difference between straight and bent waveguide ∆s [cm]	0.24	0.14	0.07
Number of bends N	2	4	3
Straight distance between P1 and P2 [cm]	1.81	1.81	1.35
Total intensity loss between P1 and P2 [dB]	3.33	13.50	17.00
Total length between P1 and P2 [cm]	1.86	1.87	1.37
**Scattered light attenuation parameter** **S = 1.04 dB/cm**			
Total scattering losses [dB]	2.97	2.99	2.19
Total bending losses [dB]	0.36	10.51	14.81
$\alpha_{\mathrm{scat},\;360{}^{\circ}}\;[\frac{dB}{360{}^{\circ}}]$	2.51	1.01	0.50
$\alpha_{\mathrm{bend},\;360{}^{\circ}}\;[\frac{dB}{360{}^{\circ}}]$	0.54	5.25	9.87
$\alpha_{\mathrm{total},\;360{}^{\circ}}\;[\frac{dB}{360{}^{\circ}}]$	3.73	6.43	10.71
**Scattered light attenuation parameter** **S = 0.78 dB/cm**			
Total scattering losses [dB]	1.45	1.46	1.07
Total bending losses [dB]	1.88	12.04	15.93
$\alpha_{\mathrm{scat},\;360{}^{\circ}}\;[\frac{dB}{360{}^{\circ}}]$	1.22	0.49	0.25
$\alpha_{\mathrm{bend},\;360{}^{\circ}}\;[\frac{dB}{360{}^{\circ}}]$	2.83	6.02	10.62
$\alpha_{\mathrm{total},\;360{}^{\circ}}\;[\frac{dB}{360{}^{\circ}}]$	4.05	6.51	10.87
**Scattered light attenuation parameter** **S = 0.30 dB/cm**			
Total scattering losses [dB]	0.56	0.57	0.42
Total bending losses [dB]	2.77	12.93	16.59
$\alpha_{\mathrm{scat},\;360{}^{\circ}}\;[\frac{dB}{360{}^{\circ}}]$	0.48	0.19	0.10
$\alpha_{\mathrm{bend},\;360{}^{\circ}}\;[\frac{dB}{360{}^{\circ}}]$	4.15	6.47	11.06
$\alpha_{\mathrm{total},\;360{}^{\circ}}\;[\frac{dB}{360{}^{\circ}}]$	4.63	6.66	11.15

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
