# Peer review of "On the Fabrication and Characterization of Polymer-Based Waveguide Probes for Use in Future Optical Cochlear Implants"

_materials, 2022, doi:10.3390/ma16010106_

Round 1
Reviewer 1 Report
This paper reports a 6" wafer level micromachining process for the fabrication of flexible waveguide probes made of PMMA as cladding and SU-8 as core material. Some comments.
1. Introduction: improve with some literature in additional from what the proposed probes can be used like on physical and biosensing: IEEE Sensors Journal 18 (20), 8381-8388, 2018; Optics express 28 (13), 19740-19749, 2020.
2. How about the reproducibility of each type of probe? And in terms of repeatability after different cycles of tests?
3. The scattering attenuation parameter was determined to be (0.7±0.3) dB/cm. What are the optimal dimension to use if was biosensor? is it possible to bring it for this topic?
4. Can we inscribe Bragg grating in the core part? Please consider to add some words about this possibility.
5. In fig. 3c we can control the cladding on the top? or we can minimize this layer till where?
Author Response
Thank you very much for your carefully reading and the comments you gave us for enhancing the paper!
Point 1: Introduction: improve with some literature in additional from what the proposedprobes can be used like on physical and biosensing: IEEE Sensors Journal 18(20), 8381-8388, 2018; Optics express 28 (13), 19740-19749, 2020.
Response 1: Thank you very much for your literature recommendations. We would like to incorporate them into the introduction in the following way:
“In addition waveguide probes can be used in a wide field of sensor applications. For example waveguide probes with inscribed Bragg gratings embedded in 3-D printed ABS sensing pads for force measurements were developed [Leal-Junior] as well as sensors for environmental monitoring and drinking water quality control based on a gold coated tilted fiber Bragg grating with immobilized Acinetobacter sp. Bacteria on the gold surface used as receptors [Cai].”
Leal-Junior: “Leal-Junior, A.G.; Marques C. FBG-Embedded 3-D Printed ABS Sensing Pads: The Impact of Infill Density on Sensitivity and Dynamic Range in Force Sensors. IEEE Sensors Journal 2018, 18(20), 8381-8388.”
Cai: “Cai, S.; Pan, H. Selective detection of cadmium ions using plasmonic optical fiber gratings functionalized with bacteria. Optics Express 2020, 28(13), 19740-19749.”
Point 2: How about the reproducibility of each type of probe? And in terms of repeatability after different cycles of tests?
Response 2: Above Figure 6 you can read that the attenuation parameters are reproducible for other probes on the same wafer. In addition there were no significant differences between the structures over the whole wafer. Even for the identical processed waveguide probes which are arranged 90° clockwise on the 6" wafer (see Figure 1 and Figure 4a). Unfortunately we can not prove this at the moment but a new process run will start soon. By using a new automated microscope for defect detection we will be able to evaluate the reproducibility of the probes better.
We don’t know exactly if we understand question 2 correctly. Is your question how the attenuation parameters change after a degradation test? At the moment we can not answer this question accurately because we assume the decrease in output power of the SBGs to be mainly caused by geometrical changes (i.e. a decrease in the incoupling efficiency) and not by a degradation of the probe. But this is a very good point, thank you for this. To evaluate the effect of probe degradation on the attenuation parameter the following procedure would be helpful:
First light is coupled into the waveguide and the scattered light curve is recorded. Afterwards the probe is aged for example by storing the probe in saline solution for a certain period of time. Then again light is coupled into the same waveguide and the scattered light curve is recorded. The attenuation parameters are extracted from the two scattered light curves and compared to each other. In theory the attenuation parameter of the aged probe should be higher than the one of the initial probe.
Point 3: The scattering attenuation parameter was determined to be (0.7±0.3) dB/cm.What are the optimal dimension to use if was biosensor? is it possible to bring it for this topic?
Response 3: Also for the use in biosensors the losses should be rather low. Anyway the losses in polymer waveguides exceed the ones in glass waveguides (dB/cm vs dB/km).
Some examples of losses in SU-8 waveguides are given here:
- 76 dB/cm: “Wang, X.-B.; Sun, J. Thermal UV treatment on SU-8 polymer for integrated optics. Optical Materials Express 2014, 4(3).”
- 8 dB/cm: “Sun, X.; Xie, Y. Effect of film compatibility on electro-optic properties of dye doped polymer DR1/SU-8. Applied Surface Science 2013, 285(B), 469-476.”
- 2 dB/cm: “Nordstrom, M.; Zauner, D.A. Single-Mode Waveguides With SU-8 Polymer Core and Cladding for MOEMS Applications. Journal of Lightwave Technology 2017, 25(5), 1284-1289.”
- 4 dB/cm: “Alt, M.T.; Fiedler, E. Let There Be Light-Optoprobes for Neural Implants. Proceedings of the IEEE 2016, 105(1), 101-138.”
Point 4: Can we inscribe Bragg grating in the core part? Please consider to add somewords about this possibility.
Response 4: The waveguides presented here are multimode waveguides. Singlemode waveguides are required for really useful Bragg Gratings.
Point 5: In fig. 3c we can control the cladding on the top? Or we can minimize this layer till where?
Response 5: The thickness of the upper cladding is 2.5 µm. Due to a high contrast in defractive indices of core and cladding material a cladding thickness of 2.5 µm is sufficient because the evanescent wave only invades about 2 µm into the cladding of the waveguide. For singlemode waveguides a cladding thickness of about 50 µm would be required because the evanescent wave invades very deep into the cladding. Furthermore a thickness of 2.5 µm for the upper cladding makes it less challenging to spincoat the cladding on top of the substrate with already processed SU-8 core layers regarding topography and the 6" waferlevel process.
Reviewer 2 Report
About the production and characterization of polymer-based 2 that study of waveguide probes for use in future optical cochlear implants is a very interesting research topic.
The study examines the fabrication and characterization of polymer-based waveguide probes for use in optical cochlear implants.
Cochlear implants are devices designed to bypass damaged sensory hair cells in the cochlea and provide stimulation directly to the auditory nerve. Research on this subject continues from different angles. A new perspective is presented in this study.
Optical waveguide-based implants are intended to be polymer-based, flexible and highly mobile.
A study on the wavy level generation of flexible polymer-based waveguides is presented. In addition, in the characterized sample, it may be possible to eliminate some chronic defects by studying the surface roughness. Each of these faults has the potential to disrupt the light directing properties of waveguides. The method used to determine faults and optical loss mechanisms differs from similar studies. The analytical approach and imaging methods used and morphological approaches have been revealed.
Considering the problem situation - method and results of the results, they can be considered as valid and reliable data that can be accessed. A full comparison of results can only be confirmed by repetition of the experiment.
The number of references can be increased and more up-to-date references should be used. Self-reference level is acceptable.
The presentation materials presented are appropriately clear and informative.
Author Response
Thank you very much for your carefully reading and the comments you gave us for enhancing the paper.
Point 1: Extensive editing of English language and style required
Response 1: the whole manuscript will be controlled by at least two native speaker again to check the English language and style.
Point 2: The number of references can be increased and more up-to-date references should be used. Self-reference level is acceptable.
Response 12: Thank you for your feedback. We would like to introduce the following sentence into the introduction of the paper to address your point:
“Waveguides exhibit bending [Marcatili, Marcuse, Ebeling] and scattering [Lacey, Payne] losses, which are experimentally investigated in this paper.”
Marcatili: “Marcatili, E.A. Bends in optical dielectric guides. The Bell System Technical Journal 1969, 48(7), 2103-2132.”
Marcuse: “Marcuse, D. Influence of curvature on the losses of doubly clad fibers. 1982, Applied Optics, 21(23), 4208-4213.”
Ebeling: "Ebeling, K.J. Integrated Optoelectronics; Publisher: Springer Berlin Heidelberg, Germany, 1993."
Lacey & Payne: “Lacey, J.P.; Payne, F.P. Radiation loss from planar waveguides with random wall imperfections. IEE Proceedings J (Optoelectronics) 1990, 137(4), 282-289.”
Payne & Lacey: “Payne, F.P.; Lacey, J.P. A theoretical analysis of scattering loss from planar optical waveguides. Optical and Quantum Electronics 1994, 26(10), 977-986.”
Reviewer 3 Report
In my opinion, the main contribution of the given article is a new approach to different types of probes for use in future optical cochlear implants based on PMMA/SU-8/PMMA. The team of authors also discusses the problems connected with the usage of designed types of probes. All information in the article is well arranged with a clear relation to the theoretical and practical part of the evaluation. Generally, the article is well prepared and subsequently provides a reader with all the necessary information about the solved task. Literature references are sufficient and from quality sources.
The abstract of the paper is appropriate and adequate. The target of the paper is unambiguous. English is good, in my opinion.
From the article's plagiarism point of view, this article seems to be prepared carefully and well. The value of similarity with other publications is low (please see attachment).
Remarks:
On the other hand, the quality of the figures is low. Therefore, I would recommend changing all of them. I would use figures with better readability and higher resolution.
Could you please provide more details about the aspherical lens?
In the case of Fig. 13 provide values of the R factors for each of the curves!

Author Response
Thank you for this positive feedback.
Point 1: On the other hand, the quality of the figures is low. Therefore, I would recommend changing all of them. I would use figures with better readability and higher resolution.
Response 1: Thank you for your feedback. We checked our uploaded word-file of the manuscript and it seems that during pdf creation the image quality was reduced to images with very low resolution. We will check this very precisely and will do some changes e.g. for Figure 1 (there are too many details and therefore it’s not clear) and the others Figures will be adjusted, too (e.g. Figure 2: format errors of “white boxes” during pdf-converting / Figure 4b: white boxes as background of the description).
Point 2: Could you please provide more details about the aspherical lens?
Response 2: Yes, of course, we can. The used asphere has a focal length of 11 mm and a NA of 0.26. The mounted lens is purchased at Thorlabs and hast the part number A220TM-A. The working distance is 6.91 mm and the lens surface is AR coated for the spectral range from 350 nm to 700 nm.
Point 3: In the case of Fig. 13 provide values of the R factors for each of the curves!
Response 3: The R squared values of the fits are the following:
SBG006: R² = 0.94628
SBG008: R² = 0.72605
SBG013: R² = 0.99142
SBG014: R² = 0.96180
Round 2
Reviewer 1 Report
The reference list is not well since there are some like the new one as ref. 28 that there is not included.
Author Response
Dear Reviewer,
thank You again for your feedback and sorry for the mistake with the not correct set references in the reference list. We checked this and inserted some more references for the use of polymer waveguides within the introduction.
Thank You and we apologize for the inconvenience.